# A Standardized Guide to Developing an Online Grocery Store for Testing Nutrition-Related Policies and Interventions in an Online Setting

**DOI:** 10.3390/ijerph18094527

**Published:** 2021-04-24

**Authors:** Pasquale E. Rummo, Isabella Higgins, Christina Chauvenet, Annamaria Vesely, Lindsay M. Jaacks, Lindsey Taillie

**Affiliations:** 1Department of Population Health, New York University Grossman School of Medicine, New York, NY 10016, USA; 2Carolina Population Center, University of North Carolina at Chapel Hill, Chapel Hill, NC 27599, USA; ihiggins@email.unc.edu (I.H.); avesely@email.unc.edu (A.V.); taillie@unc.edu (L.T.); 3Prevention Research Center, University of South Carolina, Columbia, SC 29208, USA; cchauvenet@sc.edu; 4Global Academy of Agriculture and Food Security, University of Edinburgh, Midlothian EH25 9RG, UK; Lindsay.jaacks@ed.ac.uk; 5Department of Nutrition, University of North Carolina at Chapel Hill, Chapel Hill, NC 27599, USA

**Keywords:** food policy, experimental design, red meat, fruits and vegetables, sustainable diet, food labels, food marketing and promotion

## Abstract

Simulated online grocery store platforms are innovative tools for studying nutrition-related policies and point-of-selection/point-of-purchase interventions in online retail settings, yet there is no clear guidance on how to develop these platforms for experimental research. Thus, we created a standardized guide for the development of an online grocery store, including a detailed description of (1) methods for acquiring and cleaning online grocery store data, and (2) how to design a two-dimensional online grocery store experimental platform. We provide guidance on how to address product categorization, product order/sorting and product details, including how to identify outliers and conflicting nutritional information and methods for standardizing prices. We also provide details regarding our process of “tagging” food items that can be leveraged by future studies examining policies and point-of-selection/point-of-purchase interventions targeting red and processed meat and fruits and vegetables. We experienced several challenges, including obtaining accurate and up-to-date product information and images, and accounting for the presence of store-brand products. Regardless, the methodology described herein will enable researchers to examine the effects of a wide array of nutrition-related policies and interventions on food purchasing behaviors in online retail settings, and can be used as a template for reporting procedures in future research.

## 1. Introduction

Diets rich in energy, added sugars, processed meat and unprocessed red meat, and low in whole grains, nuts, fruits and vegetables, are positively associated with risk of poor cardio-metabolic health and some cancers [1,2,3,4]. A higher-quality diet is associated with a lower risk of obesity, diabetes, cardiovascular disease and some cancers [4], emphasizing the importance of a healthy eating pattern in chronic disease prevention. However, very few adults in the US consume the recommended amounts of fruit, vegetables and red meat, potentially due to cost, food and taste preferences and cultural norms [5,6,7,8]. Nutrition-related policies and interventions that affect food retail, particularly at the point-of-selection (where consumers make decisions about what to buy) and point-of-purchase (where the purchasing decision actually occurs), are important tools for reducing the risk of nutrition-related diseases. Specific strategies include, for example, financial incentives and disincentives (e.g., discounts and taxes), warning labels, product placement and healthy default foods and beverages (e.g., pre-set menu options) [9,10,11]. In more recent years, interest in these practices as strategies to reduce the environmental harm caused by food production, such as greenhouse gas emissions [12], has also grown.

Online food retail has grown quickly in the past few years, with a marked increase due to the COVID-19 pandemic [13,14]. This sudden shift provides a unique opportunity to leverage evidence-based strategies to encourage healthy food purchases (e.g., fruits and vegetables) and discourage less healthy food purchases (e.g., red meat and sugar-sweetened beverages) in online food retail settings. Compared to brick-and-mortar stores, it is easier to modify the design of online environments to influence consumer decision-making, such as modifying prices, editing front-of-package labeling and adding default shopping cart items. With support from software developers, behavioral health and nutrition researchers can evaluate the impact of nutrition-related policies and interventions on consumer behavior in online store experiments.

Researchers in several countries (e.g., New Zealand, Singapore, UK, USA), for example, have collaborated with software developers to build a platform for online behavioral research studies, which has the appearance and functionality of a real grocery store website, including browsing, search, product pages, shopping cart and checkout [15,16,17,18,19,20,21]. These experimental online stores are also linked to food databases, with nutritional information collected from food product labels. In addition to two-dimensional online stores, researchers have also created three-dimensional “virtual” supermarkets and other online retail stores (e.g., clothing) [22,23,24,25], which are designed to mimic a real shopping experience in a brick-and-mortar store. These “virtual” online stores are built to test the impact of various retail strategies, such as food swaps, price changes and nutrition labels, on purchase intentions. However, to our knowledge, there are no detailed methods on how to create two-dimensional online grocery stores for experimental research, which is important for consistency in the literature and maximizing generalizability by creating realistic shopping experiences.

To fill this gap, we sought to create standardized methods for the development of a grocery store designed to mimic an online shopping experience for the study of nutrition-related policies and interventions in an online retail setting, including a detailed description of (1) methods for acquiring and cleaning online grocery store data, and (2) how to design a two-dimensional online store for experimental research, including product categorization, product order and sorting, product images, product details (e.g., price, nutritional information) and tagging specific products for nutrition-related interventions, using fruits and vegetables and red meat as two examples.

## 2. Materials and Methods

### 2.1. Online Grocery Store Experimental Platform

To create an online grocery store, we used an online shopping platform called Gorilla (https://gorilla.sc) developed by UK-based Cauldron Science, Ltd. (http://cauldron.sc), a company with expertise in designing platforms for online behavioral research studies. Gorilla includes a tool called the Shop Builder, which enables researchers to create an online store that can be modified according to the researchers’ goals. The Gorilla platform also provides the functionality to randomize participants to conditions and administer surveys and tasks. The platform works with third-party panel vendors to recruit participants (e.g., CloudResearch). After providing informed consent, eligible participants are directed to the online store to complete specified shopping tasks, with instructions tailored to experimental conditions (e.g., budget amount, target products). The experimental platform allows for a variety of experimental stimuli, including warning labels, taxes, discounts, default shopping cart items and suggested swaps for specific products. Patients or the public were not involved in the design, conduct, reporting or dissemination plans of our guide.

### 2.2. Online Grocery Store Data

#### 2.2.1. Data Source

To obtain online grocery store data, we contracted a company that provides custom web-scraping solutions (Marquee Data), which enabled us to compile a large dataset (<20,000 observations) in a timely manner (two weeks) at a low cost (<$3000). A similar approach has been utilized previously by other researchers (18). The data were scraped from one of the largest online grocery stores in the USA in August 2020. We selected this store because it has a large presence throughout the USA and it has affordable prices. The product data included: department, aisle (i.e., department sub-categories), shelf (i.e., aisle sub-categories), product name, price, serving size, servings per container, calories, fat, saturated fat, trans fat, carbohydrates, sugar, sodium, fiber, protein, and iron, ingredients list, UPC, shelf rank and the product images (Appendix A). The store brand name was removed from product names to minimize the influence of brand loyalty, but the products themselves were retained to provide products at a range of prices, as store brands are often cheaper. The department (broadest category), aisle and shelf (most granular category) were used by the store to designate the location of a product, and the same product could be located in multiple departments, aisles and shelves. The serving size, servings per container, calories, fat, saturated fat, trans fat, carbohydrates, sugar, sodium, fiber, protein and iron were scraped from the nutritional facts panel (NFP) data displayed on the product’s page. We also scraped “shelf rank”, which indicates a product’s order on a shelf when sorted by best seller.

We chose to scrape data at a single time-point from a single zip code because (1) it was cost-efficient, (2) it ensured all study participants were exposed to the same dataset, (3) it was too time-consuming to clean multiple datasets and (4) it ensured that the relative prices of the products were consistent for all participants. We chose to scrape data from zip code 63119 (St. Louis, MO, USA) because it represents to 95.3% the US cost-of-living (i.e., 4.7% below the average cost-of-living in the US), [26] and retailers have selected St. Louis as a pilot location for food products (e.g., the Impossible Whopper at Burger King) [27]. The initial dataset had 23,314 products, including duplicate products appearing in more than one shelf category.

#### 2.2.2. Data Cleaning

To ensure the scraped data were of high quality and to prepare the data for our planned experiments, we conducted several phases of data cleaning, including (1) defining the initial list of products and filling in missing NFP information, (2) performing a data audit, (3) tagging products and (4) online grocery store review.

### 2.3. Phase 1: Define Initial List of Products and Fill in Missing NFP Information

In the first phase of data cleaning, we decided which products to eliminate from the dataset and addressed missing nutritional information. We removed all alcoholic beverages as alcohol laws vary from state to state and some states do not allow the sale of any alcohol in grocery stores. We removed all non-edible products, such as cake toppers and cupcake liners, because our planned experiments are focused on modifying food and beverage purchases. In addition, we removed the shelves “New in Pantry” and “New in Fresh Bakery” because all products were duplicates. After these exclusions, approximately 12% of the 20,320 remaining products did not contain calorie information.

As a first step, we eliminated freshly-made and bakery products missing calorie information because nutritional information was not available on the grocery store’s website or other sources. We also deleted products missing calorie information if they had a shelf rank of greater than 60 because our store only displayed 60 products per page and we did not expect participants to browse more than one page per shelf. We did not, however, eliminate any products missing calorie information corresponding to the food groups in our planned research studies (i.e., the “Fruits & Vegetables,” “Meat & Seafood” or “Deli” departments).

For the remaining products, we extracted all missing nutritional information from the product’s image on the grocery store’s website. If the information was not available, we extracted data from the product manufacturer’s website. For fresh produce, we collected missing nutritional information from the FoodData Central website, an integrated, research-focused data system that provides data on nutrients and other food components [28]. We searched for items using as much detail as possible and selected the produce item that was the closest match, including product name, variety (e.g., “Granny Smith”), method of preparation (e.g., “raw,” “sliced,” “peeled,” “in syrup,” etc.) and other details (e.g., added vitamins). If variety, method of preparation or other details were not identified in our search, we selected the generic option. FoodData Central has several data types, including “SR Legacy,” which provides nutrient and food component values that are derived from analyses, calculations and the published literature. The “Food and Nutrient Database for Dietary Studies (FNDDS)” contains data on nutrient and food component values derived from the National Health and Nutrition Examination Survey. “Branded Foods” provides values for nutrients in branded and private-label foods from a public-private partnership. Because fresh fruits and vegetables were not branded, we selected “Survey (FNDDS)” or “SR Legacy” items instead of “Branded”, when possible. If the produce serving size was missing, we referenced the American Heart Association’s serving size recommendations [29].

### 2.4. Phase 2: Data Audit

In the second phase, we completed a data audit to identify potential systematic issues with the dataset. Four members of the study staff each reviewed 100 randomly-selected products from the dataset (n = 400 total products) and assessed the extent to which the information from the scraped data matched the information on the grocery store’s website. We found that NFP informational text from the grocery store’s website often differed from the NFP data in the image from the product manufacturer. We decided to keep the NFP information from the grocery store’s website because transcribing data from >10,000 images would be time-consuming and error-prone. To exclude advertisements and NFP images, we decided to only use the first image scraped for each product (which was always a front-of-package image). To identify extreme values of price, calories, sugar, sodium and saturated fat, we reviewed the top 1% of values for each variable within each department, and the bottom 1% for price values. We did not review the bottom 1% of values for the other variables as it is common for products to have low or zero values for calories, sugar, sodium and saturated fat. To identify outliers, we first compared the price and nutritional information of products in our dataset to the grocery store’s website. If the prices matched but still appeared extreme, we used the price from the website of a different large online grocery store. If the prices were different, we updated the value using the other store’s price. If the values for other variables matched but still appeared extreme, we used the values from the NFP image or the product manufacturer’s website, if more plausible. In total, the price, calories, sugar, sodium and/or saturated fat of 1031 products (5%) were modified and 140 products (<1%) were removed; our final dataset included 20,180 products (Table 1).

### 2.5. Phase 3: Online Grocery Store Review

We uploaded the final dataset to the online grocery store experimental platform and reviewed the first page of each shelf (Figure 1). First, we reviewed the first five products on the product page of each shelf and confirmed that the product name, image and ingredients list corresponded to the same/correct product. We also searched for incorrect applications of our point-of-selection strategies, image errors and missing and/or inaccurate product information. During our review, we found few errors regarding the application of the point-of-selection strategies, product name, image and ingredients list (<0.5%). However, we found that the servings per container and/or serving size were missing for several products. To mitigate this issue, we reviewed the departments, aisles and shelves with products relating to our planned studies. If the serving size and/or servings per container were missing, we extracted all missing nutritional information from the NFP on the grocery store’s website, or, if not available, from the NFP image. If the information was missing from both sources, we searched the FoodData Central website for missing fruit and vegetable product data and the product manufacturers’ websites for other product data [28]. In total, we updated the NFP for 814 (4%) products.

In addition to data cleaning, we “tagged” products that we wished to modify in our planned experiments. Tagging is accomplished on the “back end” of the server (versus the “front end” or user interface). The goals of two of our planned studies using the online grocery store were to (1) assess the impact of taxes and health and environmental warning labels on consumers’ decisions to purchase red meat, and (2) examine the extent to which an economic incentive and a behavioral “nudge” increased fruit and vegetable purchases among low-income adults in the US. To prepare the dataset for these experiments, we tagged relevant products with a “0” or “1” for pre-specified intervention manipulation variables. See Appendix A for details of the tagging procedures.

After tagging all products, three staff members reviewed 200 randomly-selected products (n = 600 total products) to ensure the tagging was accurate, and did not identify errors in product tagging. In total, 1718 (9%) products were tagged as eligible for the red meat tax and health and environment warning labels, including 407 (55%) products in the “Deli” department and 395 (55%) products in the “Meat & Seafood” department. Among the red meat products, 59% contained beef, 67% contained pork, 26% contained beef and pork, 0.2% contained other red meat and 91% were processed. In total, 831 (4%) products were tagged as eligible for the fruit and vegetable discount, including 538 (59%) products in the “Fruits and Vegetables” department and 71 (10%) products in the “Organic Shop” department. Among all products tagged as eligible for the fruit and vegetable intervention, 14%, 16% and 71% were frozen, canned and fresh produce, respectively.

### 2.6. Online Grocery Store Design

To maximize the external validity of our experimental findings, our goal was to make online shopping tasks as realistic as possible. To accomplish this goal, we created our own store name, “Lola’s,” and worked with a graphic designer to create a unique logo (Figure 1). In the future, we plan to develop additional graphic images to augment the store’s realistic appearance (e.g., ad banners). To mimic the appearance and functionality of a typical US grocery store’s website, we organized products by department, aisle and shelf (Appendix A). The department names are listed across the top of the screen, below the store logo. The aisle names appear when a department is clicked on, and, subsequently, shelf names appear when an aisle is clicked on. Then the participant can select a shelf to view individual products, with a maximum of 60 products per page, which are sorted according to the “shelf rank” (i.e., the popularity of the item), as scraped from the online retailer. Participants can also view products by typing words into the search bar, which will display individual products containing the inputted word(s). When a participant clicks on a product, the product appears on a new webpage, where a participant can view the product’s name and price, a front-of-package image, NFP text and an ingredient list (Figure 2).

The online grocery store has a shopping basket so participants can make multiple selections. The shopping basket shows the quantity of the product selected and plus and negative signs, which allow the participant to add or remove products from their basket. The basket also shows the cost of each product and the total cost of all products. The platform allows researchers to require that participants have a minimum and/or maximum number of items in their basket before checking out. The platform also has an optional shopping list feature, which enables researchers to outline which items participants should select from the online grocery store (i.e., task-based shopping). The shopping list includes check-boxes next to each item so participants can keep track of products in their basket and which products they still need to select, if desired. The shopping basket and shopping list are anchored to the right side of the screen (Figure 1; Appendix A).

To discover potential issues with our initial design of the store, we recruited six participants and instructed them to conduct a household shop with a budget of $50 while ‘thinking out loud.’ Based on participants’ feedback, we worked with the software developer to increase the upload time of the website, make the placement and size of the search bar more prominent, and update the “Add to basket” button to better convey when a product was added to a participant’s shopping cart.

## 3. Discussion

Through this guide, we are building the capacity to examine the impact of nutrition-related policies and interventions on food purchasing behaviors in online retail settings. Though several researchers have used online grocery stores to test healthy eating interventions [15,16,17,18,19,20,21], this guide is the first to provide a comprehensive description of how to create an online grocery store platform, which will minimize differences in the experimental platform of studies testing similar research questions (e.g., number and placement of items). Specifically, we provide guidance on how and where to obtain nutritional information for products missing such data (e.g., fresh fruits and vegetables), how to identify outliers and conflicting information and methods for standardizing the price of food items. Though the data and software used to develop our online store are proprietary, researchers will be able to mimic our approach by obtaining a contract and licensing fee from Marquee and Cauldron, respectively. In addition, we provided specific details regarding our process of tagging food items that can be leveraged by future studies examining nutrition-related policies and interventions targeting red and processed meat and fruit and vegetable purchases. Importantly, the methodology described herein can be used as a template for reporting procedures in future research, including procedures for data collection and cleaning, data sources and platform design.

We experienced several challenges in the process of obtaining product information and product images for our online grocery store. Because we used a commercial vendor to acquire data from a real US grocery store, it was not feasible to develop a “living” dataset with up-to-date products. Thus, our data reflects a single point in time, which is potentially problematic because manufacturers are constantly creating new products (e.g., meat substitutes) and reformulating the ingredients in pre-existing products. However, our results remain internally valid, given the randomization of study participants, and our goal is to update the products prior to initiating new experimental research. Another challenge was accounting for the presence of store-brand products. For example, we decided to remove store-brand names from the product name but did not create our own store-brand name; this was to avoid potential effects of brand loyalty on purchases, which may be higher for online purchasing [30]. This may reduce the generalizability of our results for a segment of the population, given the role of brand loyalty in guiding some purchase decisions. Yet, the exclusion of store-brand products means our store has wider appeal for all participants, who shop at a variety of stores, and our database includes similar products at several price points, so participants have a large breadth of options. We also identified discrepancies between the NFP text provided by the online grocery store and the image provided by the product manufacturer. We decided to use the NFP text provided by the online grocery store because it was feasible to scrape such data (vs. the NFP image provided by the manufacturer) and thus was less prone to human error, but it is possible that these data did not reflect the most up-to-date nutritional information.

A key strength of our approach is its adaptability. Though we designed the online grocery store to be compatible with our specific research goals, we also made decisions to maximize its flexibility, such as filling in missing information for all products and nutrients. This flexibility will allow us and other researchers to test a wide variety of strategies, including point-of-selection (e.g., product labels, price changes) and point-of-purchase (e.g., shopping cart defaults), on a wide variety of outcomes (e.g., dollars spent, calories purchased, proportion of items accepted); this also allows us to tailor our products, including the sample size (>20,000 products), to the study goals, upload times and for other considerations. Our decision to acquire data from a specific grocery store in a specific zip code minimized potential issues related to the availability of products sold by third-party vendors, discrepancies in list price versus offer price, and differences in prices by region. As such, though prices will not reflect the cost-of-living in the area where study participants shop for groceries, any differences in prices between our experimental store and a real store will be equal across experimental conditions. We also combined secondary data collection procedures with primary data collection efforts, which enabled us to include products that lack NFP data in our online grocery store (e.g., fresh fruits and vegetables) and solicit participants’ feedback during the design phase. However, it was not feasible to identify and correct extreme values for all nutrients, and it was not possible to add personalized features linked to user details. The latter is an important limitation because our experimental platform will not be capable of mimicking unique features of online grocery stores, such as the promotion of products based on previous purchases. For this reason, it will be important to conduct validation studies to compare consumers’ food and beverage purchases in simulated versus real online grocery stores, including participants with varying levels of digital literacy and various food preferences. Lastly, a key weakness of our guidance is the proprietary nature of our data and our license to use the experimental online grocery store platform, which may prevent other researchers from exactly mimicking our store’s products and the design of our store.

## 4. Conclusions

By following and adhering to *a priori* methods for creating online grocery stores—such as our approach to scraping and cleaning data, obtaining missing nutritional information and standardizing price data—public health researchers can maximize the comparability and reproducibility of their online experimental research. This will facilitate future meta-analyses of effect estimates stemming from this work. The need for a template will only grow as more and more consumers in the US are turning to online grocery shopping, especially as national emergencies (e.g., virus outbreaks, climate change) engender new patterns of grocery shopping. Our guidance and methods are also purposefully adaptable, allowing for flexibility in design and capacity in future studies, which is important as the form and function of online grocery stores change over time and between regions/countries: the online grocery shopping experience will continue to evolve and experimental research platforms will need to be continuously updated to remain realistic. Other steps for strengthening the generalizability of our proposed and future research include developing additional graphic images to make the shopping experience more realistic (e.g., ad banners), and improving the search function of the online grocery store (e.g., growing our list of unique search terms). In sum, the dissemination of our methods and the findings from our proposed research goals will be used to inform nutrition-related policies and interventions designed to promote healthier food shopping behaviors in online grocery stores.

## Figures and Tables

**Figure 1 ijerph-18-04527-f001:**
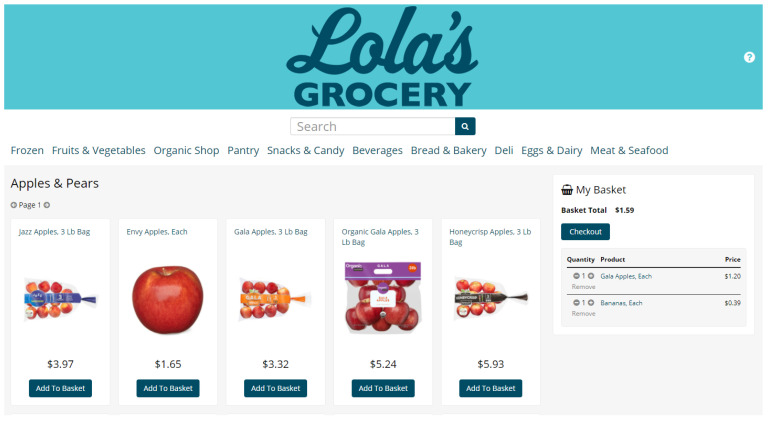
Online grocery store appearance.

**Figure 2 ijerph-18-04527-f002:**
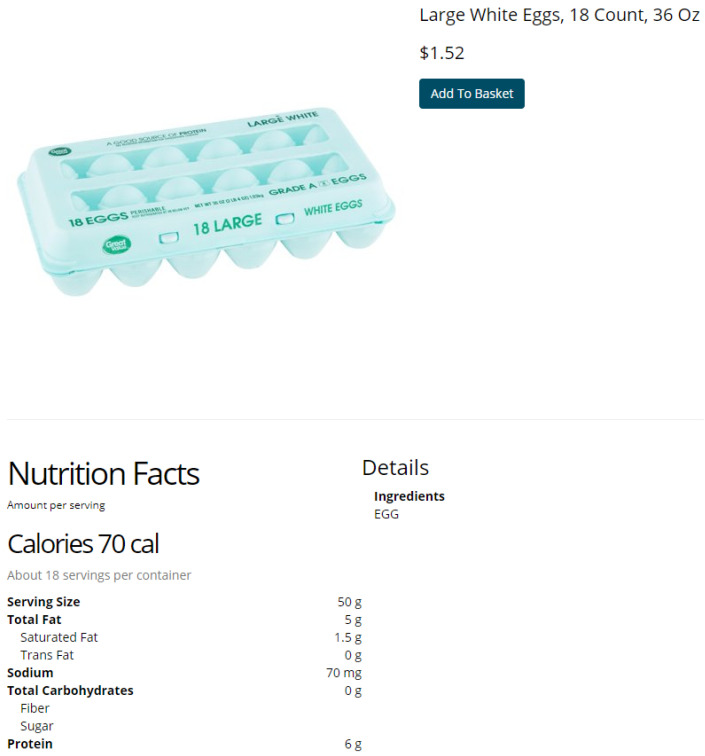
Individual product page on online grocery store.

**Table 1 ijerph-18-04527-t001:** N (%) of products by department.

Department	Product [N (%)]
Beverages	2793 (13.8%)
Bread & Bakery	954 (4.7%)
Deli	740 (3.7%)
Eggs & Dairy	2004 (9.9%)
Frozen	2192 (10.9%)
Fruits & Vegetables	914 (4.5%)
Meat & Seafood	719 (3.6%)
Organic Shop	743 (3.7%)
Pantry	6264 (31.0%)
Snacks & Candy	2857 (14.2%)
**Total**	**20,180**

## Data Availability

Data may be obtained from a third party and are not publicly available.

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
