# Peer review of "A Standardized Guide to Developing an Online Grocery Store for Testing Nutrition-Related Policies and Interventions in an Online Setting"

_ijerph, 2021, doi:10.3390/ijerph18094527_

Round 1
Reviewer 1 Report
I think the manuscript is very interesting, and that it is indeed relevant for researchers using online supermarket settings in their experiments to employ more systematic methods in designing their supermarket environments. Below some points on which I think the manuscript could be improved.
Introduction:
I think the authors provide a good introduction, covering the relevant topics and highlighting the relevance of the manuscript. In the last paragraphs, they mention 2 and 3-D online supermarket environments, but then it is not made explicit whether both such environments will be addressed, or only 2 or only 3-D environments. This should be made more explicit. This information would also be good to add explicitly to the abstract, as well as the method.
One important question that to my opinion needs to be addressed, is why the researchers did not include some form of evaluation of the supermarket design? For example user experience evaluation, or invite potential participants to utilize the supermarket or utilize expert panels in the different phases of designing the supermarket?
Method and materials:
Line 117: It says: because it represents 95.3% the US cost-of-living. It is not completely clear to me what this sentence means. Do the authors mean that this zip code is representative for 95.3% of the US population with respect to the cost-of-living? How is this calculated?
Line 178: We uploaded the final dataset into the online grocery store experimental platform and reviewed the first page of each shelf. And line 213: Then the participant can select a shelf to view 213 individual products, with a maximum of 60 products per page. I wonder here, how the order of the products on the pages is defined?
Line 146: it says: We searched for items using as much detail as possible and selected the produce item that was the closest match.: it is not clear to me what is meant by ´the item that was the closest match’.
Line 148: It says: We included varietal and method of preparation in our search. What is meant by this exactly and how was this applied in the search?
Line 149: What is meant by Survey FNDDS or SR legacy here?
Line 174 : It says: In total, 140 products (<1%) were removed and our final 174 dataset included 20,180 products (Table 1). Could it also be described for how many of the products the information was adjusted based on reviewing the previously mentioned values (by comparing to other websites, to the manufacturers info, etc)?
It would be helpful to provide some images of the online grocery format a bit earlier onwards, in the beginning of the method section, as it remains rather abstract what this format offered by Gorilla.sc looks like exactly.
Results:
General note: It is unclear to me why these results are presented as results in this manuscript, as the manuscript to my understanding mainly is about the design of the supermarket and how to go about designing such a supermarket for research. Hence, it is not very clear to me what I as a reader can do with these results or how they are useful with respect to the supermarket design? If it is useful with respect to the supermarket design for other researchers: then I would suggest to add some explanation as to how these percentages are informative for researchers wishing to implement such a supermarket in their research as well.
Discussion:
Line 262: Why is it that this choice was made, was it somehow verified which information source was actually correct? Would it not be better to utilize the information source that is verified as being correct?
As this is presented as an effort to also help other researchers interested in employing online supermarkets in their research, I would suggest that the authors would include a short paragraph that more explicitly adresses how other researchers can benefit from this manuscript and how they exactly can employ what was described in this manuscript in their own research. For example, can certain scripts be used easily and adjusted according to other researchers’ wishes and study goals? Will these scripts and the supermarket design be made available for other researchers to use?
Line 290: How exactly will this advance the science regarding financial incentives and disincentives, warning labels, product placement, and healthy default food and beverage options? I doubt whether the proposed experimental studies should be addressed here, as this is not the focus of the manuscript.
Reviewer 2 Report
While this manuscript has an interesting title, I am doubting whether sufficient rigor has been applied. The introduction is well written, but does not elaborate on the argument that a research-setting (in this case an mock online grocery store) should just be as close to the real world as possible (a real online grocery store). Online grocery stores are developing rapidly (in functionality, appearance, etc.), so the guidance proposed will become outdated soon.
Science proceeds on a plane of abstraction. While the study provides relevant insights into how this research team built an online mock grocery stores, it is unclear how this work contributes to future methodological challenges.
Overall, I think substantial changes should be made to improve this manuscript. The authors should not just make the changes suggested, but also improve the overall quality of the work.
Some overall points to consider:
- For the introduction section, readability could be improved by removing unnecessary details and revising the structure. Generally, start with the overall scope of the topic and clearly identify the problem that you will focus on in this manuscript. Give a clear indication of the theoretical framework you use, and the gap in the literature that your research will fill. Lastly, concretely list the contributions that your paper will make to the state-of-the-art in the field.
- For the discussion section, consider your study's limitations better and focus on both the theoretical and practical (public health) implications of your work.
- Although my comments are not exhaustive, please see below a number of points that may help to improve the study design and manuscript.
Abstract ant title:
L2-3 (title): Suggest to change the title to: “Testing Public Nutrition Policies in an Online Setting: A Standardized Guide to Develop Online Grocery Stores.
L13-15: Your title suggests that the guide is particularly for online grocery stores that help research public policies for nutrition. If that is the case, please include it in the background.
L13-15: Not sure if I agree with the scope; are you suggesting that simulated online grocery store platforms are innovative tools to study PoS and PoP interventions in general, or PoS and PoP interventions in online grocery stores? As a tool to study interventions that might affect shopping behavior in brick-and-mortar grocery stores, they have more limitations than when used as a tool to study interventions in online grocery stores. Is the guide in this manuscript useful for both purposes?
L15-18: You are not describing the methods here, but the results.
L23-25: This did not seem part of your study’s background. What is the relevance?
L25-27: This is not a conclusion.
Materials and methods:
L78-81: Could you provide background on how the Gorilla platform compares to other platforms?
L101: Suggest to explain the terms “aisle” and “shelf” in an online grocery setting
L161-162: Reference?
Reviewer 3 Report
The article is well written and focused, and conforms to aims and scope of the journal. The English language is appropriate and understandable, only minor spell check being required. Also, the methods are described with sufficient details to allow another researcher to reproduce the protocol. The manuscript addresses an interesting topic and prior to publication some minor corrections are . required. In this sense, in the final conclusions section should be shorter and pointed main reached goals of the research. The results area is very limited, and there is no added table/figure to support the results. Moreover, the citation style and font should be changed according to the journal requirements.
Round 2
Reviewer 2 Report
Thank you for addressing most of the suggestions made. Unfortunately, I still have my doubts whether sufficient rigor has been applied.
While the study provides relevant insights into how this research team built an online mock grocery stores, the "flight level" of the discussion is - in my opinion - not meriting scientific publication.
Author Response
We thank the reviewer for providing their comments.